## [Decision Letter · Decision Letter 0]

15 Dec 2025

Dear Dr. Rebeck,

Thank you for submitting your manuscript to PLOS ONE. After careful consideration, we feel that it has merit but does not fully meet PLOS ONE’s publication criteria as it currently stands. Therefore, we invite you to submit a revised version of the manuscript that addresses the points raised during the review process.

We look forward to receiving your revised manuscript.

Kind regards,

Ioannis Liampas, MD. PhD

Academic Editor

PLOS One

**Journal Requirements:**

“This work was supported by NIH grants R01AG071745 (GWR), F30AG082448 (NL), and R35CA283926 (JM), and by the Alzheimer’s Association grant 24-1310437 (HP). Dr. Mandelblatt was also supported by the Frank M. Ewing Foundation Endowed Chair of Hematology and Oncology. Funding support also came from the Georgetown University Medical Center.”

4. In the online submission form you indicate that your data is not available for proprietary reasons and have provided a contact point for accessing this data. Please note that your current contact point is a co-author on this manuscript. According to our Data Policy, the contact point must not be an author on the manuscript and must be an institutional contact, ideally not an individual. Please revise your data statement to a non-author institutional point of contact, such as a data access or ethics committee, and send this to us via return email. Please also include contact information for the third party organization, and please include the full citation of where the data can be found.

5. Please include captions for your Supporting Information files at the end of your manuscript, and update any in-text citations to match accordingly. Please see our Supporting Information guidelines for more information: http://journals.plos.org/plosone/s/supporting-information ..

Reviewers' comments:

Reviewer's Responses to Questions

**Comments to the Author**

1. Is the manuscript technically sound, and do the data support the conclusions?

Reviewer #1: Yes

Reviewer #2: Partly

2. Has the statistical analysis been performed appropriately and rigorously?

Reviewer #1: Yes

Reviewer #2: N/A

3. Have the authors made all data underlying the findings in their manuscript fully available?

Reviewer #1: Yes

Reviewer #2: Yes

4. Is the manuscript presented in an intelligible fashion and written in standard English?

Reviewer #1: Yes

Reviewer #2: Yes

Reviewer #1: The manuscript presents a relevant topic with potential scientific value; however, several issues require substantial revision to improve the quality of the manuscript.

• The abstract reports changes in sERSCs and sIPSCs, PV firing rates, and inactivation block, but provides no numerical value, confidence intervals, or effect sizes. Phrases such as “caused increases” and “significantly lowering firing rates” should be supported with actual measurements.

• Overinterpretation of the conclusion in the abstract section for an ex vivo electrophysiology.

• The introduction lacks a clear problem statement and does not particularly highlight the importance of this topic.

• Some methodological steps, such as instrument settings and replicates, lack detail.

• Statistical methodology description is lacking information about the tests applied to compare groups, and how multiple comparisons were corrected.

• Ethical approval statements can be improved by adding the committee reference number ID or the protocol number.

• The introduction section lacks a critical literature context, citing older references without integrating the latest mechanistic insights.

• A graphical workflow is needed to summarise the experimental sequence (genotyping → doxorubicin → slice prep → patch clamp).

• For reproducibility, osmolarity, pH, and pipette resistance criteria should be added to the patch clamp internal solution

• Typographical errors are present throughout the manuscript, such as in the discussion section PVinactivtion needs space, and eitherGABA or AMPA is missing space.

• Provide clearer relevance on developmental neurophysiology examples as they are not directly linked to ex vivo outcomes.

• Strengths and limitations of the study are not acknowledged.

• Future research direction is not discussed

• Conclusion of the manuscript was not mentioned in the end of the manuscript.

• Journal formatting guidelines (PLOS ONE style) are not consistently followed.

• Some citations are incorrectly formatted.

Reviewer #2: This manuscript examines the impact of the APOE4 allele, a major genetic risk factor for Alzheimer’s disease (AD) on synaptic and circuit function in the entorhinal cortex (EC), both at baseline and in response to the chemotherapeutic agent doxorubicin. They used humanized APOE knock-in mice and employed whole-cell patch clamp electrophysiology to measure excitatory and inhibitory neurotransmission in layer 2/3 pyramidal neurons and parvalbumin-positive (PV) interneurons. They examined both baseline genotype effects and responses to doxorubicin, a chemotherapy agent used as an acute stressor to model cancer-related cognitive impairment. The authors conclude that APOE4 impairs circuit flexibility by disrupting PV interneuron function and blocking adaptive responses to stress, potentially explaining heightened vulnerability to cognitive decline in APOE4 carriers facing Alzheimer's pathology or chemotherapy exposure. While the work addresses an important question linking APOE4 to circuit-level dysfunction, but few methodological concerns interpretative issues limit the significance of the study.

The authors use different APOE knock-in mouse models across experiments (Sullivan et al. 1997 model in Figures 1-2 vs. Foley et al. 2022 model in Figures 3-4) without adequate justification. While they acknowledge this limitation, it undermines comparisons between baseline genotype effects and doxorubicin responses.

Several key findings rely on modest sample size of animals. For example, in Figure 4A APOE4 vehicle group has only 1 mouse. This makes it impossible to determine whether non-significant findings represent true null effects or insufficient statistical power. This is especially problematic for interpreting the lack of doxorubicin effects in APOE4 mice, which is central to the manuscript's conclusions.

This study addresses important questions about APOE4-related circuit dysfunction and responses to stress, with potential relevance to both Alzheimer's disease and chemotherapy-related cognitive impairment. However, significant methodological limitations particularly the use of different mouse models, small sample sizes, incomplete mechanistic investigation, and questionable clinical relevance of the doxorubicin paradigm limit the impact of the findings. With revisions, this work could make a valuable contribution to understanding APOE4 effects on neural circuits.

**Do you want your identity to be public for this peer review?** For information about this choice, including consent withdrawal, please see our For information about this choice, including consent withdrawal, please see our Privacy Policy .

Reviewer #1: No

Reviewer #2: No

---

## [Author Response · Author response to Decision Letter 1]

2 Feb 2026

We appreciate that the reviewers felt that our manuscript “presents a relevant topic with potential scientific value” and “could make a valuable contribution to understanding APOE4 effects on neural circuits.” We agree that the comments and our responses throughout the manuscript have improved the work; we thank the reviewers for their time and insights.

Reviewer #1:

• The abstract reports changes in sERSCs and sIPSCs, PV firing rates, and inactivation block, but provides no numerical value, confidence intervals, or effect sizes. Phrases such as “caused increases” and “significantly lowering firing rates” should be supported with actual measurements.

For space considerations in the Abstract, we could not add as many values as we would like. We now added p-values to support our statement about the effects of APOE genotype on spontaneous excitatory and inhibitory post-synaptic current amplitude. We included actual measurements and confidence intervals for the E/I ratios showing that APOE4 is associated with a higher E/I ratio. Finally, we included the percentages of PV cells that displayed inactivation block. Additional details about important statistical measures are in the text and figure legends.

• Overinterpretation of the conclusion in the abstract section for an ex vivo electrophysiology.

We edited the Abstract to include our use of ex vivo analyses and removed the last statement about impaired circuit flexibility in APOE4 to narrow the scope. These changes better reflect our data.

• The introduction lacks a clear problem statement and does not particularly highlight the importance of this topic.

We have consolidated portions of the Introduction and dedicated a specific paragraph to address the unknowns that our study seeks. Among other changes, we expanded some previous research on the effects of APOE genotype on cognitive impairment induced by cancer chemotherapies.

• Some methodological steps, such as instrument settings and replicates, lack detail.

We went through our methods and believe that the instrument settings are included. For replicates, we have tried to be more transparent in in the numbers of mice and cells used per condition. When possible, we simplified in the Figure Legends the numbers of cells and mice used per data panel to make the sample sizes more apparent. We also added to the Methods that several brain slices from the same mice were used in the electrophysiology experiments to increase the number of cells that could be analyzed to address our research questions.

• Statistical methodology description is lacking information about the tests applied to compare groups, and how multiple comparisons were corrected.

We have included any post-hoc comparisons tests in our Methods section. We used Bonferonni correction for correction of the alpha-value in the Mann-Whitney U test. Tukey's HSD has inbuilt corrections for multiple comparisons via the studentized range statistic "q" which adjusts the significance threshold for all comparisons.

• Ethical approval statements can be improved by adding the committee reference number ID or the protocol number.

We have included the IAUCUC protocol number.

• The introduction section lacks a critical literature context, citing older references without integrating the latest mechanistic insights.

We have expanded the introduction to consider more and newer findings about the effects of APOE and chemotherapy on neuronal functioning. Throughout the manuscript, we have added nineteen new references to incorporate more relevant literature and ideas. We think these changes have helped justify the experiments that we have undertaken.

• A graphical workflow is needed to summarise the experimental sequence (genotyping → doxorubicin → slice prep → patch clamp).

We have added a graphical workflow as a new Figure 1 that we believe will help readers understand our experiments. We have adjusted the other Figure numbers accordingly.

• For reproducibility, osmolarity, pH, and pipette resistance criteria should be added to the patch clamp internal solution.

We have added the internal Osm and pH to the description of the patch clamp internal solution, which also contains the pipette resistance.

• Typographical errors are present throughout the manuscript, such as in the discussion section PVinactivtion needs space, and eitherGABA or AMPA is missing space.

These typographical errors have been fixed.

• Provide clearer relevance on developmental neurophysiology examples as they are not directly linked to ex vivo outcomes.

We added our consideration of the choice to address our questions in an ex vivo model to the Introduction. We have added a sentence in the Discussion about the relevance of studies in young mice to cognitive impairment related to APOE and chemotherapy with aging.

• Strengths and limitations of the study are not acknowledged.

We have added a paragraph discussing the strengths and weaknesses of our study.

• Future research direction is not discussed

A statement of directions has been added to the final paragraph of the Discussion section.

• Conclusion of the manuscript was not mentioned in the end of the manuscript.

We have edited the conclusion and added it under its own heading after the Discussion.

• Journal formatting guidelines (PLOS ONE style) are not consistently followed.

We have tried to follow guidelines throughout, making several changes from our original submission. We will work with editors to correct any inconsistencies that we have missed.

• Some citations are incorrectly formatted.

We have reformatted the manuscript in Vancouver style. There is one paper that is available ahead of print and one that is in BioRxiv, and we would update those as needed.

Reviewer #2: This manuscript examines the impact of the APOE4 allele, a major genetic risk factor for Alzheimer’s disease (AD) on synaptic and circuit function in the entorhinal cortex (EC), both at baseline and in response to the chemotherapeutic agent doxorubicin. They used humanized APOE knock-in mice and employed whole-cell patch clamp electrophysiology to measure excitatory and inhibitory neurotransmission in layer 2/3 pyramidal neurons and parvalbumin-positive (PV) interneurons. They examined both baseline genotype effects and responses to doxorubicin, a chemotherapy agent used as an acute stressor to model cancer-related cognitive impairment. The authors conclude that APOE4 impairs circuit flexibility by disrupting PV interneuron function and blocking adaptive responses to stress, potentially explaining heightened vulnerability to cognitive decline in APOE4 carriers facing Alzheimer's pathology or chemotherapy exposure. While the work addresses an important question linking APOE4 to circuit-level dysfunction, but few methodological concerns interpretative issues limit the significance of the study.

The authors use different APOE knock-in mouse models across experiments (Sullivan et al. 1997 model in Figures 1-2 vs. Foley et al. 2022 model in Figures 3-4) without adequate justification. While they acknowledge this limitation, it undermines comparisons between baseline genotype effects and doxorubicin responses.

The development of a new human APOE knock-in model in 2022 gave us the opportunity to transition to using mice with less “genetic drift” from those developed in 1997. We have now included better justification for this change, including our reference about direct comparisons of these two mouse models, which largely share phenotypes. We also include sentences in the Results and in the Discussion to further identify this limitation and explain our justification.

Several key findings rely on modest sample size of animals. For example, in Figure 4A APOE4 vehicle group has only 1 mouse. This makes it impossible to determine whether non-significant findings represent true null effects or insufficient statistical power. This is especially problematic for interpreting the lack of doxorubicin effects in APOE4 mice, which is central to the manuscript's conclusions.

We regret having used some data that was insufficient; we removed the comparisons that depended on single mice from the manuscript. In particular, we eliminated Figure 4A, which concerned counts of parvalbumin-positive cells. These data were not important to our main conclusions; they only demonstrated that the chemotherapy did not change the number of PV-positive cells in the EC, which was not expected from a short treatment of our type. Other issues about sample sizes and statistical power are considered above in response to Reviewer 1.

This study addresses important questions about APOE4-related circuit dysfunction and responses to stress, with potential relevance to both Alzheimer's disease and chemotherapy-related cognitive impairment. However, significant methodological limitations particularly the use of different mouse models, small sample sizes, incomplete mechanistic investigation, and questionable clinical relevance of the doxorubicin paradigm limit the impact of the findings.

As noted above, we have addressed some of these concerns in this revision and added a section about the limitations of the work, to better put into context the importance of the findings.

---

## [Editor Report · Decision Letter 1]

3 Feb 2026

APOE4 and doxorubicin impair inhibitory interneuron function and homeostatic regulation in the entorhinal cortex

PONE-D-25-62615R1

Dear Dr. Rebeck,

We’re pleased to inform you that your manuscript has been judged scientifically suitable for publication and will be formally accepted for publication once it meets all outstanding technical requirements.

Kind regards,

Ioannis Liampas, MD. PhD

Academic Editor

PLOS One

---

## [Editor Report · Acceptance letter]

PONE-D-25-62615R1

PLOS One

Dear Dr. Rebeck,

I'm pleased to inform you that your manuscript has been deemed suitable for publication in PLOS One. Congratulations! Your manuscript is now being handed over to our production team.

Kind regards,

on behalf of

Dr. Ioannis Liampas

Academic Editor

PLOS One